# The Frontiers of Nanomaterials (SnS, PbS and CuS) for Dye-Sensitized Solar Cell Applications: An Exciting New Infrared Material

**DOI:** 10.3390/molecules24234223

**Published:** 2019-11-20

**Authors:** Edson L. Meyer, Johannes Z. Mbese, Mojeed A. Agoro

**Affiliations:** 1Department of Chemistry, University of Fort Hare, Private Bag X1314, Alice 5700, South Africa; emeyer@ufh.ac.za; 2Fort Hare Institute of Technology, University of Fort Hare, Private Bag X1314, Alice 5700, South Africa

**Keywords:** nanomaterials, metals sulphides, synthesis, near-infrared materials, DSSCs

## Abstract

To date, extensive studies have been done on solar cells on how to harness the unpleasant climatic condition for the binary benefits of renewable energy sources and potential energy solutions. Photovoltaic (PV) is considered as, not only as the future of humanity’s source of green energy, but also as a reliable solution to the energy crisis due to its sustainability, abundance, easy fabrication, cost-friendly and environmentally hazard-free nature. PV is grouped into first, second and third-generation cells. Dye-sensitized solar cells (DSSCs), classified as third-generation PV, have gained more ground in recent times. This is linked to their transparency, high efficiency, shape, being cost-friendly and flexibility of colour. However, further improvement of DSSCs by quantum dot sensitized solar cells (QDSSCs) has increased their efficiency through the use of semiconducting materials, such as quantum dots (QDs), as sensitizers. This has paved way for the fabrication of semiconducting QDs to replace the ideal DSSCs with quantum dot sensitized solar cells (QDSSCs). Moreover, there are no absolute photosensitizers that can cover all the infrared spectrum, the infusion of QD metal sulphides with better absorption could serve as a breakthrough. Metal sulphides, such as PbS, SnS and CuS QDs could be used as photosensitizers due to their strong near infrared (NIR) absorption properties. A few great dependable and reproducible routes to synthesize better QD size have attained much ground in the past and of late. The injection of these QD materials, which display (NIR) absorption with localized surface plasmon resonances (SPR), due to self-doped p-type carriers and photocatalytic activity could enhance the performance of the solar cell. This review will be focused on QDs in solar cell applications, the recent advances in the synthesis method, their stability, and long term prospects of QDSSCs efficiency.

## 1. Introduction

Energy production, distribution and sustainability have become one of the pressing challenges in underdeveloped countries. This is because of the low power output from the generating dams due to changes in climatic conditions. The political and geometric challenges of oil and gas power plants, nuclear power stations and coal-fired power plants are experienced globally. This has resulted in a binary advantage for photovoltaic application as a solution to the energy crisis and climate change [1,2]. Solar energy is one of the most promising and sustainable photovoltaic sources of power due to their abundance. Photovoltaic (PV) is classified as a green energy source, which is environmentally hazard-free, cost-friendly, and can easily be fabricated. Their market growth of 40% over two decades has made them the fastest-growing energy innovation [3]. 

Solar photovoltaic uses the photoelectric effect as direct sunlight conversion into electricity, such as silicon/thin-film solar cells or through photochemical effect, such as quantum dot sensitized solar cells (QDSSCs) [4,5]. The use of QDSSCs over other types of solar cells is linked to their tunable bandgap, size-dependence, high extinction coefficients, ease of fabrication and good energy conversion efficiency with multiple excitations [6,7,8,9,10]. Research on QDSSCs, as third-generation solar cells with promising prospects, has increased because of the increase in the number of research outputs (as seen in Figure 1) on QDSSCs year by year.

According to Shockley–Queisser [11], the first-generation solar cell with 33% efficiency and the second-generation solar cell with 54% efficiency has paved the way for third-generation solar cell thin-film technology to a better stage of commercialization in PV technology [12,13,14]. Third-generation solar cells using near-infrared (NIR) material quantum dots, such as PbS, CuS and SnS, could play a vital role in increasing conversion efficiency. This will replace the low spectral response, which is one of the major shortfalls of the ideal DSSCs, by possessing a better-oxidized electron at the LUMO level into the conduction band, as well as better regeneration by HUMO to their oxidized form [15,16]. In a typical build-up of DSSCs, QDSSCs portrait a similar architecture with distinctive characteristics like several microns thick semiconductor photoanode (ZnO or TiO2), photosensitizers, such as QDs (CuS, SnS, PbS), CE and electrolyte with redox couple as displayed in Figure 2. The working principle of QDSSCs is as follows: Upon the absorption of light, QDs photosensitizer absorbed on the TiO2 to generate photo-excited electrons at the interface of the conduction band (CB) and QDs. Followed by the oxidation reaction of the oxidize electron from the QDs at the (CB) and the holes synchronously oxidize the electrolyte by a reduction reaction, which is quickly migrated to the external circuit through the conductive substrate and the electrolyte interfaces thereby creating an electric current. The oxidized electrolyte is recycled back by donating oxidized electrons through the cycling circuit in the cell. Semiconductor materials, such as CuS, SnS, PbS, are commonly used to fabricate quantum dots (QDs) [6,17]. The QD photosensitizer has various features in the working mechanism and structure of photovoltaic cells; the robustness of the cell is increased, through the control of its size due to their tunable bandgap, easy fabrication and near-infrared properties [17]. One of the major challenges to good size-dependence of QDs is the pathways to synthesized size QDs [18,19,20,21,22,23]. This study will highlight the recent fabrication route, the advances in QDs and their potential in QDSSCs applications, as shown in Figure 2.

## 2. Recent Advances in SnS, CuS and PbS QDs Synthesis

The fabrication of commercialized film in optoelectronic and solar cell applications using the quantum dot near-infrared is an in-demand area of research. Semiconductors IV–VI series like PbS, CuS and SnS with low narrow bandgaps have been highlighted as a current hotspot in energy conversion research (see Table 1). Their easy reproducibility, cost-friendliness, stability and promising performances have made researchers attracted to these semiconductors [24]. The adoption of PbS, CuS and SnS in solar cell applications is due to their novel surface properties, shape, size, composition and mono-disparity. In addition, several overviews on the recent synthesis and application of quantum dot near-infrared have been investigated. Therefore, a great emphasis on the methods that give better uniform quantum dot materials is needed, such as single-source precursors [25], hydrothermals [26], template-directed approaches [27], hot-injection [28], etc.

Alivisatos et al. synthesized copper diethyldithiocarbamate using oleic acid and dodecanethiol via the mixed solvent. They obtained Cu_2_S nanocrystal from ammonium and Cu(II) acetylacetonate as their starting materials through the injection reaction [39]. Application of this fabricated nanocrystal with CdS in photovoltaics yielded 1.6% energy conversion efficiency on plastic and glass substrate with a longer stability. Zhao et al. revealed Cu_2-x_S of different nanocrystal compositions from (covellite) CuS to Cu_1.97_S through the optimization of various synthetic techniques, such as solventless thermolysis, sonoelectrochemical and hydrothermal approaches [39]. Characteristic evaluation of these materials’ properties indicated that CuS is more stable than Cu_2_S [40]. On the other hand, solventless thermolysis in an N_2_ atmosphere was utilized to produce uniform Cu_2_S from a copper thiolate precursor, with growth from small nanoparticles to nanodisks and orthorhombic to monoclinic Cu_2_S [41]. 

Karthik et al. adopted a single-source precursor method to fabricate CuS by solution thermolysis using [Cu(SON(CNiPr_2_)_2_)_2_] 1,1,5,5-tetra isopropyl-2-thiobiuret of copper(II) complex. Various morphologies were observed, such as trigonal crystallites, hexagonal disks and spheres. This points to factors like the concentration of the precursor, reaction temperature and growth time. On the other hand, the used of oleylamine in the thermolysis approach gives rise to Cu_7_S_4_ nanoparticles as a blend of anilite (orthorhombic) and roxbyite (monoclinic) phases (see Figure 3). The injection of octadecene solution precursor into hot oleylamine produces pure anilite Cu_7_S_4_ nanoparticles, while the injection of oleylamine solution precursor into hot dodecanethiol resulted in djurleite Cu_1.94_S nanoparticles. Both anilite and djurleite phases produce an indirect bandgap in the optical properties of the materials [42].

Various synthetic routes are employed to fabricate SnS quantum dots. Hickey et al. adopted thioacetamide through the synthetic route of hot injection with oleylamine and oleic acid, octadecene, Sn[N(SiMe_3_)_2_] and trioctylphoshine to fabricate SnS nonocrystals [39]. To control the shape of nanoparticles, the ratio of oleylamine/oleic acid was altered. A similar approach was used to fabricate SnS by Liu et al. through the injection of S(SiMe_3_)_2_ solutions with octadecene into SnCl_2_ solution in oleylamine [43]. The control of the size distribution was achieved through changing synthesis thermal condition; this produced the orthorhombic phase at all temperatures. Also, the colloidal approach of the turnable size of SnS is well detailed. They used an injection of the precursor, such as sulfur-oleylamine with Sn-oleylamine into hexamethyldisilazane (HMDS) at different temperatures. This produces SnS nanocrystals of a direct bandgap of 1.63 to 1.68 eV. This is directly linked to the size and morphology obtained in the range of 8–60 nm and spherical shape, while the unique crystal is due to larger particle sizes [44].

One of the most appealing semiconductors of the IV–VI series are the PbS QDs, which is due to their strong quantum confinement effects and near-infrared abilities. High-quality and quantity PbS has been produced using the relatively greener, solventless and heterogeneous method [45]. They use analytical grade oleylamine and PbCl_2_ which was blended and heated with the addition of sulfur-oleylamine. The process was stopped with the addition of hexane and the precipitate was centrifuged to obtain PbS as the final product. The characterization of PbS using photoluminescence revealed a 40% quantum yield from the particles and a stokes of 10 meV. Cao et al. [46] adopted the non-coordinating solvent approach by stabilized oleylamine in a reaction involving Pb stearate and S, resulting in a high-quality PbS with strong emissions and absorption in the near-infrared area. A similar route was used through a mixture of Pb oleate and bis(trimethylsilyl)sulphide according to a study by Hines and Scholes. Their concentrations were on the particle size of the final fabricated materials after the precipitate had been removed. Their finding revealed a turnable particle size and bandgap in the near infrared region. The emission spectrum further cemented the narrow size possession of the material (as seen in Figure 4) [47].

## 3. Overview of Single-Source Molecular Precursor Synthesis of QDs

The main constraint in the fabrication of QDs that have a better size-dependence and are more mono-disperse is the synthetic route that is adopted. This has led to innovative approaches to fabricating quantum dots of control shape and size [49]. Fabrication approaches including the single-source approach, hot injection, colloidal route, hydrothermal, microwave, solvothermal, electrodeposition, aqueous solution, pyrolysis, precipitation [50,51,52,53,54,55,56,57,58,59], etc., has been adopted for fabrication QDs. Single-source precursors have been distinguished as an effective approach that produces high-quality QDs. This is owing to their shape and controlled size through the use of capping agents in the decomposition of single-source precursors at low temperatures by one solvothermal route [60]. The advantages of this technique are numerous, such as ease of handling, low-temperature growth, air and moisture stability and low toxicity. This has placed single-source precursors over conventional precursors; using one precursor reduces the possibility of contamination on the film, and this allows for intrinsic control of the film, which occurs during the pre-reaction. Also, they used ligands to reduce impurities, such as carbon in the alkyl group which is incorporated in the film, as well as offering a degree of control. However, some precursors allow the control of the phase deposited through the use of molecular cluster compounds and core structure precursor molecules [60]. Overview of the approach on the following semiconducting materials, such as PbS, CuS and SnS quantum dots will be discussed.

### 3.1. PbS Quantum Dots

The study by Bo Hou et al. adopted a synthetic path to fabricate reliable, high performance and effective QDSSCs using PbS QDs. These were achieved by altering the precursor concentration at constant temperature and time. This resulted in many various sizes with high reproducibility and narrow size distribution of the fabricated PbS QDs (Figure 5) [61]. Boadi et al. reported the decomposition of Pb(S_2_CNiBu_2_)_2_ to form PbS with various morphologies based on the temperature utilized in the fabrication processes [62]. The material thermalized at 100 °C and displaced a sphere in diameter of 6.3 nm. While cubic crystallites at a diameter of 60 nm in a mixture of 150 °C and small spherical quantum dots were produced. However, the decomposition of Pb(S_2_CNEt_2_)_2_ at 230 °C in hot dodecylamine and phenyl ether produces large cubes in a similar study by Lee et al. [63]. Boadi et al. reported that the injection of hot phenyl ether with Pb(S_2_CNEt_2_)_2_ precursor, resulted in well-defined structures and star-shaped PbS QDs [62]. 

### 3.2. CuS Quantum Dots

Copper chalcogenides have been well reported for their localized surface plasma resonances and their displaced strong NIR absorption. CuS is intermediate with strong potential in the photocatalytic application and self-doped p-type carriers due to NIR absorption [64]. In addition, different capping agents can be utilized to fabricate various morphologies of CuS QDs [65,66,67]. It is very important to look into major factors, such as the optical properties, microstructure defects, crystal growth procedures and cubic-hexagonals of CuS QDs due to ultrafast nonlinear optical properties. Looking at photovoltaic semiconductor materials, it is of great importance to understand the evolution of the optical properties from nanoparticles to QDs [68]. A recent study has shown that fabricated CuS QDs using capping agents, such as polyvinylpyrrolidone, produces particles with sizes of 2–4 nm and 5–7 nm, according to Luther et al. (as seen in Figure 6), with hexagonal and cubic phases being observed for the QDs. Furthermore, planar defects like stacking, twins and dislocations are due to a reasonable amount of CuS QDs.

Crystallographic phases suitable for crystal growth of CuS QDs are dominated by oriented attachment, while optical absorption of CuS QDs is evaluated in linear and nonlinear regimes. The presence of electron/hole traps indicated that CuS QDs displaced the lower bandgap energy [69]. 

### 3.3. SnS Quantum Dots

SnS are semiconductors in group IV–VI. They have a direct and indirect bandgap of 1 eV to 1.3eV and high coefficient absorption [70]. These absorptions were within the visible and near- infrared areas of the electromagnetic spectrum. These are the area where optimum efficiency in QDSSCs can be attained. This further underlining SnS QDs as potentials materials for photovoltaic technology. Capping agents and temperature affect the shape and size resulting in SnS within the range of 10–28 nm. The mixture of octadecylamine capped and oleic acid produce cubic and small nanocrystals using single molecular precursors on fabricated SnS QDs [71]. Deng et al. prepared SnS QDs turnable size using tinoleylamine solution into hexamethyldisilazane through the injection of the sulfur-oleylamine precursor at different heats. The smaller optical bandgap indicated larger nanocrystals [72]. 

## 4. Molecular Precursor Complexes

Molecular precursor has paved way for the synthesis of monodispersed, better crystalline materials, semiconducting quantum dots, containing metal and chalcogenide [73]. Hiroki and Mitsunobu reported the use of two molecular Cu(II) precursor solutions of EDTA and propylamine obtained from both amine and formic mixtures. The adjustment of Cu concentrations in the precursor solution of ethanolics gives rise to novel Na-free FTO films [74]. The major draw-back of molecular precursors are pinpoints to the fabrication route of the ligand by producing small yields, which constitutes a shortfall for large scale applications in various fields of research. Chemical properties of the complexes represent the second draw-back as a result of their valence orbital having been shielded [75]. Coordination compounds often resulted in different structural geometry based on properties of metal ion and ligands [76]. 

Therefore the complex precursor should be of good solubility in solvents and have quite good thermal and chemical stabilities. The use of the dithiocarbamate ligand has given rise to the synthesis of various novel molecular complexes. They have the ability to produce complexes with transition metals, in addition to stabilizing metals ion at high oxidation states. These have expanded their application in various fields including medicine, polymer photo stabilizers, biology, antioxidants, chemistry, photovoltaic, and so many more. Dithiocarbamate complexes of Sn(II), Cu(II) and Pb(II) have been utilized as molecular precursors for thin film growth [77].

### 4.1. Dithiocarbamate Cu(II) Complexes

Geraldo et al. used a bidentate complex of [Cu{S_2_CNR(CH_2_CH2OH)}_2_] in the d_x_^2^_–y_^2^ orbital state to form square planar geometry. This resulted in a distorted square planar molecule crystallised complex and dimer intermolecular Cu bond in the form of independent centrosymmetric monomers and similar solid-states [78]. Some dithiocarbamate complexes with Cu(II), such as pyrrolidinedithiocarbamate have also been reported for their strong antiviral, antioxidant and anti-inflammatory properties and they are able to transport metal through membranes [79]. Four coordinate bidentate metal ion structures within ligand and two sulphur of square planar geometries were reported by De Lima et al. [80]. Victor et al. in their study used a solvent molecule of Cu(Et_2_dtc)_2_ complex, which pointed to the axial of copper ion in the photochemical activity. The dimer coordinated with the Et2dtc• radical produced involves the reaction of initial complex and intermediate; accompanied by microseconds the dithiocarbamate radicals, which reunited to produce a tetranuclear cluster. These turned out to be challenging on stationary photolysis when looking at the mechanism of photochemical transformations of the Cu(Et2dtc)2 complex [81,82].

### 4.2. Dithiocarbamate Pb(II) Complexes

The use of Pb(S_2_CNRR)_2_ formulated as lead(II) dithiocarbamate complexes to fabricate PbS QDs through single-source precursors approach has been reported by [83]. The advantage of these over traditional organometallics is the absence of toxic metals like lead alkyls or H_2_S. They are easy to fabricate and stable for months and give high yields with thermolysis. The decomposition of Pb (S_2_CNEt_2_)_2_ complex in tri-n-octyl phosphine oxide produces cube-shaped quantum dots [82]. Similarly, Cheon et al. produced rod-based star-shapes, cubes and highly faceted and truncated octahedrons of different structures with the same precursor (Figure 7) [63]. On the other hand, spherical to cubic shapes can be produced by altering the dodecanethiol ligand ratio through single- source precursors to form PbS. Hydrolytic processes were adopted to fabricate star-shaped PbS QDs using bis(thiosemicarbazide)lead(II) [Pb(TSC)_2_Cl_2_] [84]. Cubes shapes of PbS were synthesized using ethylene diamine with Pb xanthate complex in room temperature decomposition through the greener synthetic route [85]. Pb(II) complexes of aromatic carboxylate with thiosemicarbazide or thiourea were adopted as a molecular precursor to fabricate a stable PbS QDs by aqueous or non-aqueous solvent decomposition [86]. 

### 4.3. Dithiocarbamate Sn(II) Complexes

Tin(II) and Tin(IV) species are one of the semiconductors with interesting redox chemistry and are stable in both metal–organic complexes and organometallics. Fabrication of PbS thin-films with different single-source precursors is well documented in the literature, such as Sn(S_2_CNEt_2_)_4_, Sn(SPh)_4_ and Sn(SCH_2_CH_2_S)_2_, Sn(S-cyclohexane)_4_, and Sn(SCH_2_CF3)_4_ [87,88]. Punarja et al. use Tin(II) of [Sn(S_2_CNRR′)_2_] to fabricate unsymmetrical dithiocarbamates. The thermolysis of the complexes produces absorbance in the near-infrared region with an optical bandgap of 1.2eV, good morphology and composition [89].

## 5. Recent Advances in PbS, SnS and CuS Quantum Dot Application for Dye-Sensitized Solar Cells

To replace the low efficiency in ideal DSSCs and promote their commercialization as potential photovoltaic devices, semiconductor QD materials with better light absorbers could be used as a replacement in QDSSCs. Semiconductor materials with unique optoelectronic properties, multiple exciton generation, high extinction coefficients and bandgaps with a size control less than 10 nm are classified as special QDs [90,91,92,93,94,95]. Ideal QD dyes should have many properties including producing a photoexcited electron upon the absorption of light. In 2008, Zhao et al. used recombination control to produce 8.21% efficiency in their study [96]. Ren et al. overcome the instability of QDSSCs in their study through metal oxyhydroxide coatings of photoanode to produce a 9.73% efficiency [97]. Du et al. reported efficiency above 11.39% by using carbon CE-based QDSSCs [98]. Furthermore, Jiao et al. adopted nitrogen-doped mesoporous carbons with an efficiency exceeding 12% as unique potential QD-based cells [99]. Recently, Luther’s group obtained an efficiency of 13.43%, which is currently the optimum efficiency in QDSSCs [100].

### 5.1. Recent Advances in PbS Quantum Dots 

The direct bandgap of 0.37eV with a vast exciton Bohr radius of 18 nm at room temperature has made PbS QDs one of the potential p-type semiconductors [101]. Owning to their 400–2500 nm near-infrared area and visible region through adjusting the dot size [102], Sargent et al. utilized the high-mobility hybrid perovskite by inserting PbS QDs into their matrix, which resulted in 4.9% electroluminescence power conversion efficiency [103]. Chuang et al. synthesized electron blocking/hole-extraction layer and light-absorbing layer thin-film by using PbS-EDT CQD and PbS-TBAI to construct a long-life air stable solar cell with 8.55% efficiency [104]. While Lee et al. adopted the solid-state PbS DQ-SSCs on substrate using SILAR/spiro-OMeTAD (2,2,7,7-tetrakis (N,N-di-p-methoxyphenylamine)-9,9-spirobi-fluorene) HTL/Au to achieve 1.46% [105]. Furthermore, Seok et al. constructed a spin-SILAR/PEDOT:PSS (poly(3,4-ethylene- dioxythiophene) polystyrene sulfonate)/Au/P3HT (Poly-3-hexylthiophene) with 2.0% efficiency using (bl-TiO_2_)/mp-TiO_2_/PbS QDs FTO/blocking. This later improved by using the same approach with a CH_3_NH_3_PbI_3_ (MAPbI_3_) shell resulting in 3.2% efficiency [106]. 

Recently, Im et al. used mesoscopic PbS QD-SSCs to achieve 4.96% efficiency and later improved the efficiency to 9.2% with the injection of an MAPbI_3_ interlayer [107]. Karthik et al. used the same mesoscopic PbS QDSSCs by inserting CuS in a rapid rate to the cationic exchange reaction. With the aid of Cu-oleylamine and oleic acid as the capping agents to produce 8.07% efficiency under 1 sun test condition (AM1.5G 100 mW/cm^2^) [108]. Seok et al. adopted the colloidal route to fabricate PbS QDSSCs merged with FTO/bl-TiO2/m-TiO2/multiple-layered PbS QDs/P3HT/PEDOT:PSS/Au to produce 2.9% power conversion efficiency [109]. The efficiency was later improved to 3.9% using TiO_2_ nanorod ETL/PbS QDs/P3HT HTL interface by radial directional charge transport to fabricating PbS QDSSCs [110]. Similarly, Zhang et al. achieved 3.57% efficiency by TiO_2_ nanorod array/spiro-OMeTAD/Au in FTO/compact PbS QDSSCs (as seen in Figure 8) [111].

### 5.2. Recent Advances in SnS Quantum Dots

SnS QDs displayed better ways of absorbing the whole solar spectrum due to blue shift absorption properties. Deepa and Nagaraju, in their study, fabricated SnS QDs with sizes of 2.4 to 14.4 nm through the chemical precipitation approach by controlling the growth time. The SnS QDs with different structures were fabricated with and without the buffer layer and apply in solar cell test (see Table 2). The efficiency varies when the QD dyes were loaded on the cell with the buffer layer, which reduces the efficiency and improved the fill factor (FF) [112,113]. Tang et al. gives a detailed photocatalytic activity of rhodamine B under the halogen lamp of fabricated SnS QDs using Sn(II), oleic acid, octadecene, thioacetamide, oleyamine and trioctylphosphine as the starting materials [114]. Das and Dutta reported efficient degradation of trypan blue dye photocatalyst in sunlight with the aids of mercaptoacetic acid capping agent to fabricate SnS nanorods [115]. To further improve SnS QDSSC efficiency, the limitations that hinder their large scale commercialization such as secondary phase formation and bulk defects, diffusion length, short minority carrier lifetime, band alignment, solar cell configuration, and back contact should be given important consideration to enhance the efficiency of QDSSCs

### 5.3. Recent Advances in CuS Quantum Dots

Metal sulphides, such as copper sulfides (Cu_2_S and CuS) [67], are well-known in QDSSCs as efficient materials with redox couple of polysulfide electrolytes. They have displayed good electrocatalytic activity, low resistance and strong stability with redox reaction polysulfides [67]. In addition, CuS and Cu_2_S are generally adopted in QDSSCs. The near-infrared absorption of CuS due to self-doped p-type carriers and photocatalytic activity has made them potential semiconducting materials for various applications [39]. The J–V curve of the QDSSCs was evaluated in 1 sunlight and the finding revealed a display of high efficiency for CuS 2 h-based QDSSCs with (η) of 4.27%, (FF) of 0.49, (Voc) of 0.603 V and (Jsc) of 14.31 mA cm^−2^, but are much better when compared with conventional Pt CEs ( η = 1.11%, Voc = 0.523 V, FF = 0.24 and Jsc = 8.83 mA cm^−2^). CuS 2 h-based efficiency is linked to the catalytic properties and the aid of CuS counter electrolyte low charge transfer resistance with polysulfide electrolytes (as seen in Figure 9). 

## 6. Conclusions, Further Improvement and Future Studies

Semiconductors with near-infrared regions, quantum confinement and high absorption coefficients are highly desired in photovoltaic cell research. New scientific advances and the potential of greater power conversion efficiency in these semiconductors has contributed to the noteworthy strides in the third-generation solar cells. To be a pace-setter in third generation photovoltaic technology, further advancements should be made on improving the efficiency of QDSSCs. Additionally, the emphasis on the stability of efficient electrolytes in fabricated solar cells is highly recommendable. The synthesis of absorbing QD near infra-red region materials that could absorb the whole visible light harvesting regions should be also considered. Research on the reduction of charge recombination and the fabrication of QDs that will absorb photoelectrodes with better porosity should be investigated. Looking at the numbers of research articles on semiconductor QDs and the huge challenges encountered in the improvement of QDSSCs, new breakthroughs in improving the efficiency of QDSSCs is inevitable. 

## Figures and Tables

**Figure 1 molecules-24-04223-f001:**
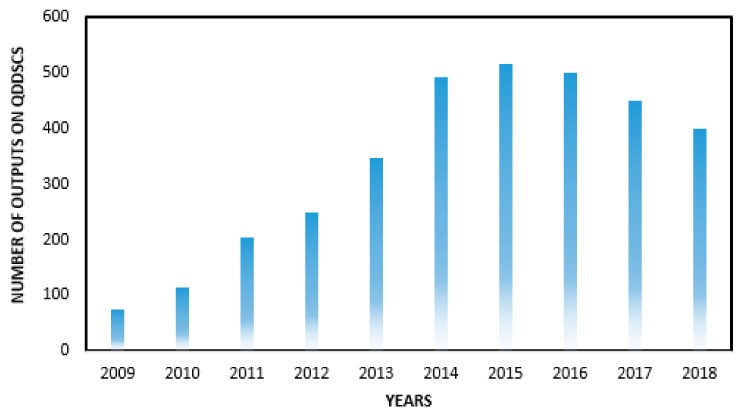
Research outputs from 2009 to 2018 on quantum dot sensitized solar cells (QDSSCs) (reproduced from ref. [6] with permission from the American Chemical Society).

**Figure 2 molecules-24-04223-f002:**
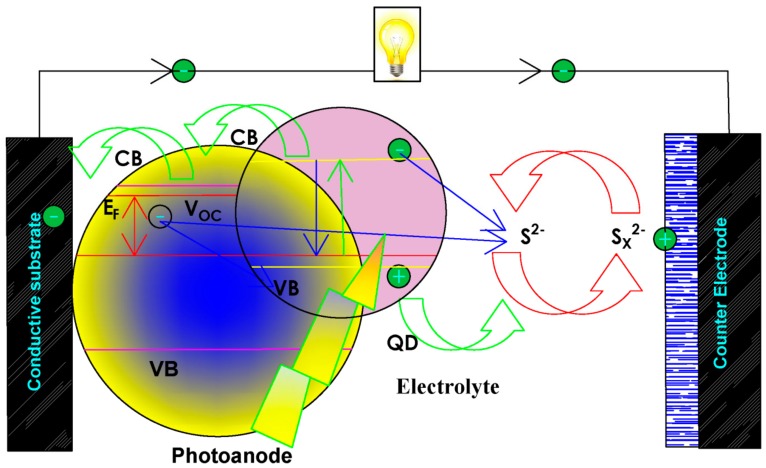
Schematic illustration of the QDSSCs working procedure involving the quantum dot (QD) photosensitizer, counter electrode, photoanode and electrolyte.

**Figure 3 molecules-24-04223-f003:**
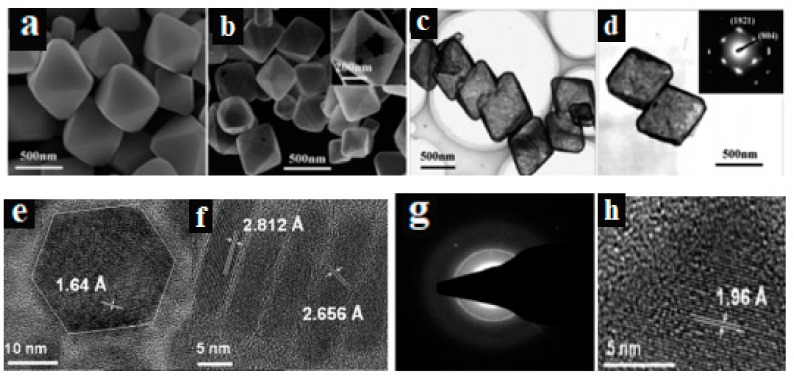
Morphology images of (**a**) octahedral Cu2O crystals, (**b**) octahedral CuxS cages, (**c**) TEM octahedral and (**d**) the SAED pattern of octahedral (reproduced from ref. [41] with permission from the American Chemical Society); (**e**,**f**) are for the sample prepared using 10 mM solution of the precursor at 200 ℃ for 1h showing flat and standing hexagonal nanodisks and (**g**) SAED. (**h**) TEM images for the sample prepared using 20 mM solution of the precursor at 280 ℃ (reproduced from ref. [42] with permission from the American Chemical Society).

**Figure 4 molecules-24-04223-f004:**
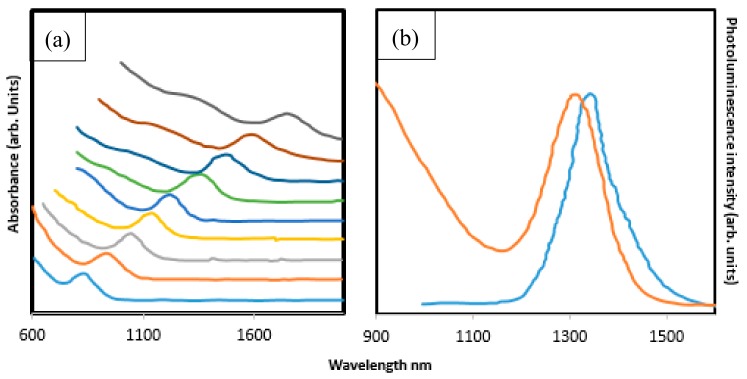
UV and PL absorption properties of PbS nanocrystals (**a**,**b**) (reproduced from ref. [48] with permission from the American Chemical Society).

**Figure 5 molecules-24-04223-f005:**
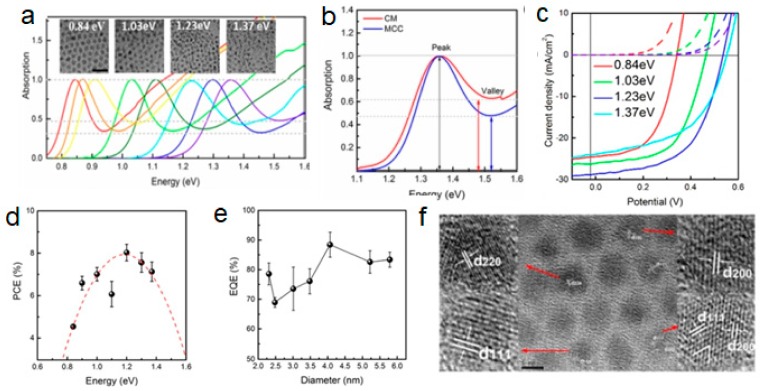
PbS QDs (**a**) absorption spectra. (**b**) Comparison of the first exciton peak. (**c**) Representative J−V curves of QDSSC with PbS QDs. (**d**) PCE values (black legend) as a function of QDs optical ε_gap_. (**e**) Peak EQE values measured at a 400 nm wavelength. (**f**) HRTEM images (reproduced from ref. [61] with permission from the American Chemical Society).

**Figure 6 molecules-24-04223-f006:**
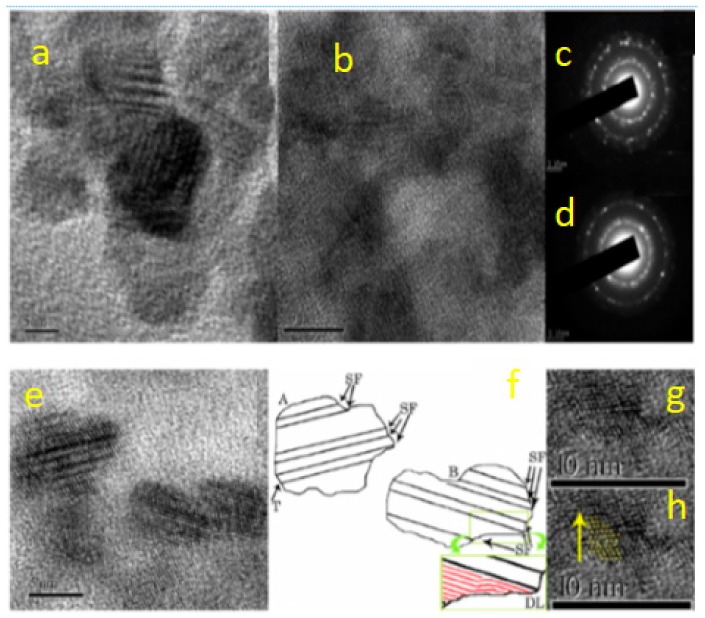
HRTEM images and SAED patterns of CuS-PVP nanoparticles (NPs) (**a**) and (**c**) and QDs (**b**) and (**d**). HRTEM image of CuS-PVP QDs showing abrupt edges (**e**). Dislocations, twins, and stacking faults are illustrated in (**f**). A twin region formed between the cubic and hexagonal phases of CuS is illustrated in (**g**) and (**h**). The stacking sequence is determined as ABCABABCAB in the direction indicated by the arrow (reproduced from ref. [69] with permission from the American Chemical Society).

**Figure 7 molecules-24-04223-f007:**
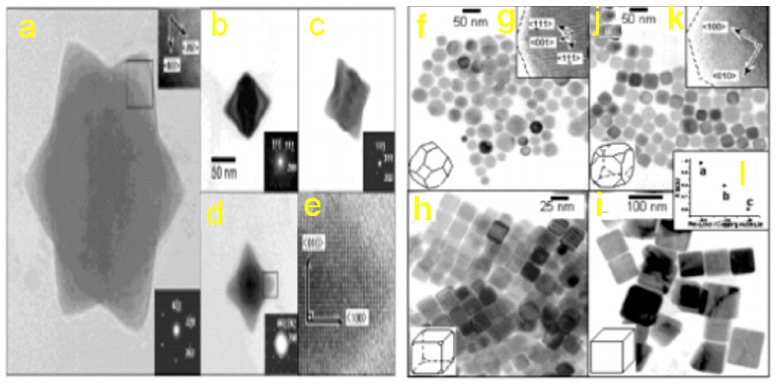
(**a**) TEM image of star-shaped PbS nanocrystals synthesized at 230 °C. (Inset) HRTEM image of lattice fringes with zone axis of (111). (**b**–**d**) TEM images and electron diffraction patterns with zone axis of (**b**) (110), (**c**) (112) and (**d**) (100), respectively. (**e**) HRTEM image of zoomed fringes with zone axis of (100). (**f**–**h**) Dodecanethiol (DT) capping molecular system. The ratio of precursor to capping molecules is (**a**) 1/400, (**b**) 1/100 and (**c**) 1/50, respectively. (**i**) Dodecylamine (DDA) capping molecular system with the ratio of precursor to capping molecule of 1/100. (**j**) HRTEM image of a round truncated octahedron with zone axis of (110), and (**k**) a truncated octahedron with zone axis of (100). (**l**) The R factor depends on the relative ratio of precursor to capping molecules (reproduced from ref. [63] with permission from the American Chemical Society).

**Figure 8 molecules-24-04223-f008:**
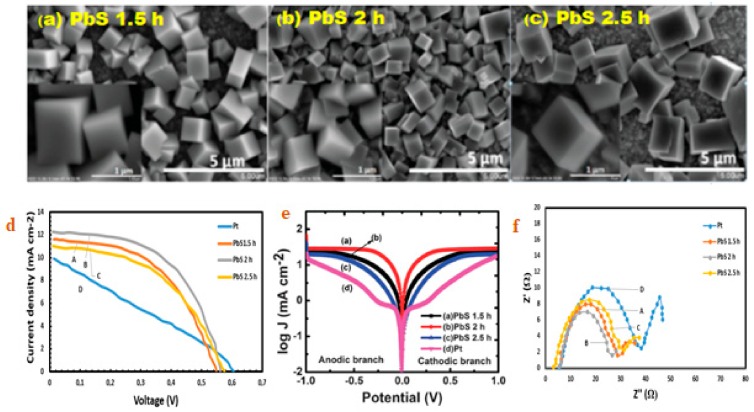
(**a**–**c**) PbS CEs SEM images, (**d**) J–V, (**e**) Tafel and (**f**) Nyquist plots (reproduced from ref. [111] with permission from the American Chemical Society).

**Figure 9 molecules-24-04223-f009:**
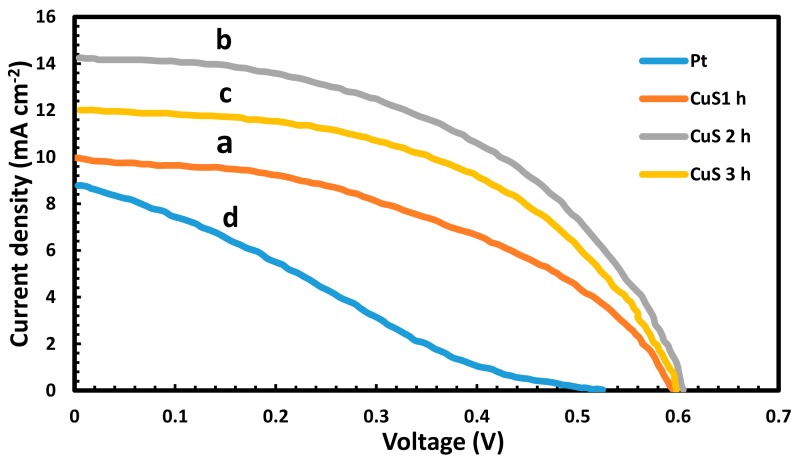
(J–V) of TiO2/CdS/CdSe/ZnS QDSSCs with CuS and Pt CEs (reproduced from ref. [130] with permission from the Elsevier).

**Table 1 molecules-24-04223-t001:** J–V properties of PbS, CuS and SnS QDSSCs with various photoanodes and CEs.

Quantum Dots	Photoanode	Electrolyte	CEs	ECE (%)	Year	Ref.
CuS	TiO_2_/CdS/CdSe/ZnS	S^2−/^S_x_^2−^	Pt	3.18	2015	[29]
Cu_2_S-CBD/N719	TiO_2_	4-tert-butylpyridine	Pt	8.34	2016	[30]
CuInSe/ZnS	TiO_2_	S^2−/^S_x_^2−^	Cu_2_S	8.10	2016	[31]
PbS QDs	TiO_2_	S^2−/^S_x_^2−^	Cu_2_S	2.24	2016	[32]
PbS/CdS/ZnS	TiO_2_	S^2−/^S_x_^2−^	Cu_2_S	7.19	2016	[33]
PbS	TiO_2_	S^2−/^S_x_^2−^	Cu_2_S	1.30	2018	[34]
PbS CQD	ZnO	TBAI	Carbon	3.6	2017	[35]
SnS QDs	TiO_2_	4-tert-butylpyridine	Pt	0.326	2015	[36]
SnS	TiO_2_	4-tert-butylpyridine	Au	0.0102	2014	[37]
SnS QDs	TiO_2_	S^2−/^S_x_^2−^	Pt	0.072	2019	[38]

**Table 2 molecules-24-04223-t002:** Photovoltaic parameters of PbS, CuS and SnS solar cells.

Cell	V_OC_ (mV)	J_SC_ (mA/cm^2^)	FF (%)	H (%)	R_S_ (kΩ cm^2^)	RSH (kΩ cm^2^)	n	J_0_ × 10^−7^ (mA/cm^2^)	Year	Ref.
CuS5/FTO5 μm	0.37	5.81	52	1.12	13.6	26.2	-	-	2017	[116]
CuS 2h	0.612	15.52	0.452	4.29	7.65	1.68	1.33	2.676	2017	[117]
CuS/NF	0.61	17.8	0.54	4.93	7.4	7.26	2.1	-	2017	[118]
CuS	0.58	0.71	48	1.38	-	-	-	-	2017	[119]
CuS	688	9.37	67	4.31	-	-	-	-	2019	[120]
PbS90	0.644	12.17	0.588	4.61	7.56	1.45	0.5	5.01	2015	[121]
ITO/PbS-TBAI/Al	0.45	17.86	52.0	4.18	-	-	-	-	2017	[122]
PbS 2 h	0.560	11.20	0.55	3.48	8.58	2.6	1.21	3.04	2015	[123]
PbS	495	17.46	0.639	5.52	-	-	-	-	2019	[124]
PbS/CuS/HNS QD	0.61	23.8	64.4	9.3	-	-	-	-	2019	[125]
PbS QD	0.56	25.01	67.3	9.43	-	-	-	-	2018	[111]
SnS	0.39	1.7	0.72	2.36	-	-	-	-	2017	[126]
SnS QDs	342	78	38.2	0.0102	-	-	-	-	2014	[44]
Mo/SnS/CdS/i-ZnO/ITO	290	17.2	56	2.8	-	-	-	-	2019	[127]
SnS-S	746	18.09	0.67	8.96	-	-	-	-	2019	[128]
FTO/πSnS/CdS/i-ZnO/ITO	113	3.40	42	0.15	-	-	-	-	2018	[129]

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
