# Peer review of "The Frontiers of Nanomaterials (SnS, PbS and CuS) for Dye-Sensitized Solar Cell Applications: An Exciting New Infrared Material"

_molecules, 2019, doi:10.3390/molecules24234223_

Round 1

Reviewer 1 Report

The manuscript by Meyer et al. is a review of PV materials based on SnS, PbS and CuS. The manuscript overview main achievements in this topic. The material is presented in a consistent way and it will be interesting for the broad audience and readers. In my opinion, the manuscript could be accepted after some minor revisions:

There are some typos in the manuscript, and even in the title (PdS instead of PbS); I would advise authors to add more recent publications (2018-2020) relating to this topic; Perhaps, there will be valuable to add a table summarizing main results/achievements.

Author Response

Good Morning Sir/Ma

Please find the attached.

Kind regards

Agoro M.A

Reviewer 2 Report

All figures taken from other sources are of very low quality. Instead of cut and paste from other sources, you should download original Figures from their work and use them. For this permission from the publishers is required. Similarly, Figure 2 that you have made needs to be improved. You should choose better quality colors so that the resolution of the Figure improves. 

Author Response

Good Morning Sir/Ma

Please find the attached

Kind regards

Agoro M.A
